# Diagonalizing Affinity Matrix to Identify Clustering Structure

## Abstract

Affinity matrix-based clustering constitutes an eminent approach within the domain of data mining. Nevertheless, prior research overlooked the opportunity to directly exploit the block-diagonal structure of the affinity matrix for the purpose of identifying cluster formations. In this paper, we propose an affinity matrix-based clustering strategy, termed as *DAM*, which employs a traversal algorithm to discern high-density clusters within the graph weighted by the affinity matrix, thereby establishing a traversal sequence. This sequence is subsequently utilized to permute the affinity matrix, thereby revealing its intrinsic block-diagonal structure. Moreover, we introduce an innovative split-and-refine algorithm that autonomously detects all diagonal blocks within the permuted matrix, ensuring theoretical optimality in the presence of well-separated clusters. Extensive evaluations on six real-world benchmark image clustering datasets demonstrate the superiority of our method over contemporary state-of-the-art clustering techniques.

## 1 Introduction

In the present era, characterized by an abundance of data, vast quantities of information are continuously amassed and stored across numerous databases, necessitating the development of sophisticated analytical techniques to extract meaningful insights Jain et al. (1999). Among such techniques, cluster analysis is instrumental in unveiling the inherent groupings or structures within datasets. Clustering algorithms, being unsupervised, exhibit remarkable versatility and are employed across diverse fields, including data analytics, computer vision, and image processing Xu & Tian (2015); Xing & Zhao (2024).

Despite the emergence of a plethora of clustering algorithms derived from various theoretical frameworks, accurately identifying clusters based on spatial data distribution remains a formidable challenge, particularly when the number, density, orientation, and shape of the clusters are undefined Fraley & Raftery (1998). Addressing these complexities necessitates the use of robust and adaptable clustering methods, capable of discerning intricate data characteristics.

Traditional clustering algorithms may be broadly categorized into four principal types Fraley & Raftery (1998): partition-based MacQueen (1967); Liu et al. (2023a); Hu et al. (2023); Mussabayev et al. (2023), hierarchical Menon et al. (2020); Huang et al. (2023), affinity matrix-based Sun & Du (2018); Dong et al. (2023); Liu et al. (2023c), and density-based methods Ding et al. (2023); Qiu & Li (2023); Ester et al. (1996). Among these, affinity matrix-based methods have garnered considerable attention in recent years, owing to affinity matrix construction advancements in convex optimization techniques and the adoption of deep neural networks Xie & Wang (2021); Taştan et al. (2023); Zhang et al. (2021); Fan et al. (2022); Liu et al. (2022); Li et al. (2023b); Kong et al. (2023); Xu et al. (2020); Liu et al. (2023b; 2020a; 2021); Zhang et al. (2019a). Data naturally tends to form distinct clusters; hence, the affinity matrix learned from the data ideally exhibits a block-diagonal structure, wherein each block represents a cluster characterized by high intra-block similarity and low inter-block similarity. Nevertheless, despite the potential utility of this structure, existing methods have predominantly focused on enhancing the structure of the affinity matrix itself, rather than thoroughly exploring the relationship between the block-diagonal structure of the affinity matrix and the resultant clustering outcomes Taştan et al. (2023). Consequently, current research lacks strategies that directly leverage the block-diagonal form of the affinity matrix to reveal the underlying clustering structure.

This paper introduces the *Diagonalizing Affinity Matrix (DAM)* clustering method. As illustrated in Fig. 1, high intra-cluster similarity and low inter-cluster similarity imply the presence of a block-diagonal form within the affinity matrix. Our approach incrementally searches for dense clusters within the graph weighted by the affinity matrix, employing a traversal algorithm that identifies high-density clusters and establishes a traversal sequence. The block-diagonal structure is subsequently realized by permuting the affinity matrix in accordance with this sequence. This density-based traversal algorithm

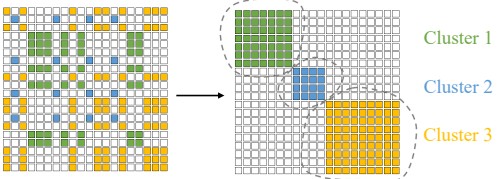

Figure 1: An illustration of the affinity matrix diagonalizing process, wherein white points signify low similarity, and colored points denote high similarity.

transforms the affinity matrix into a block-diagonal form, thereby facilitating both automatic and interactive cluster analysis, whilst enhancing comprehension of data distribution and correlations. Furthermore, we propose an innovative split-and-refine algorithm that autonomously detects all diagonal blocks within the permuted matrix by determining segmentation index that maximize the sum of elements within these blocks, ensuring theoretical optimality in instances of well-separated clusters.

Our contributions are summarized as follows:

- **Learning Diagonal Blocks for Clustering:** We introduce a strategy that exploits the potential block-diagonal structure of the affinity matrix. We associate each diagonal block with a distinct cluster, thereby identifying the clustering structure from block-diagonal representation.

- **Block-Diagonal Generation and Identification:** We develop a density-based search strategy capable of discovering clusters in the graph weighted by the affinity matrix, accommodating varying densities. The affinity matrix is permuted according to the traversal order, and a rapid block-diagonal identification method is proposed, ensuring theoretical optimality in the case of well-separated clusters.

We evaluate the performance of the proposed clustering method on six benchmark image clustering datasets, demonstrating that the proposed DAM achieves superior clustering performance compared to contemporary state-of-the-art methods.

## 2 RELATED WORKS

### 2.1 EXPLORING BLOCK-DIAGONAL STRUCTURE OF AFFINITY MATRIX IN CLUSTERING

Numerous studies have explored the block-diagonal properties of the affinity matrix for clustering. Yang et al. Yang et al. (2019) propose a joint robust multiple kernel clustering method that promotes an affinity matrix with optimal block-diagonal properties using a regulariser and self-expressiveness. Liu et al. Liu et al. (2020b) seek a block-diagonal structure by imposing a $K$-block-diagonal constraint, while Wang et al. Wang et al. (2020) enforce such structure through a non-convex regulariser. Qin et al. Qin et al. (2021) introduce a semi-supervised clustering approach that enforces the block-diagonal structure, addressing both sparsity and smoothness. Lin et al. Lin & Chen (2022) present an adaptive block-diagonal representation that maintains convexity, whereas Liu et al. Liu et al. (2022) propose an adaptive low-rank kernel block-diagonal representation, mapping the input space to a linearly separable Hilbert space. Qin et al. Qin et al. (2022) establish a theoretical link between spectral clustering and graph construction using block-diagonal representation. Xu et al. Xu et al. (2023) and Li et al. Li et al. (2023b) develop efficient block-diagonal graph learning approaches, while Kong et al. Kong et al. (2023) ensure a $k$-block-diagonal representation matrix, and Li et al. Li et al. (2023a) construct a block-diagonal similarity matrix using ordered partition points. The exploration of block-diagonal properties has also extended to multi-view clustering, as shown by Yin et al. Yin et al. (2021) and Liu et al. Liu et al. (2023b). Other matrix optimisation-based methods are discussed in Xie & Wang (2021); Taştan et al. (2023); Zhang et al. (2021); Fan et al. (2022); Liu et al. (2022); Li et al. (2023b); Kong et al. (2023). Recently, neural network-based approaches

have garnered significant attention. Xu et al. Xu et al. (2020) propose a latent block-diagonal representation model for nonlinear graph construction, while Liu et al. Liu et al. (2023b) incorporate both block-diagonal and diverse representations into a multi-view clustering network. Other notable neural network-based works include Liu et al. (2020a; 2021); Zhang et al. (2019a).

However, all these methods predominantly focus on enhancing the potential block-diagonal structure within the affinity matrix to better represent the underlying clustering structure. Subsequently, they rely on spectral clustering to identify clusters, utilizing the spectrum (i.e., eigenvalues) of the affinity matrix for dimensionality reduction, followed by the application of K-means clustering in the reduced dimensional space. These approaches neglect to directly exploit the block-diagonal structure of the affinity matrix to derive clustering results.

## 2.2 BLOCK-DIAGONAL GENERATION AND IDENTIFICATION

A variety of methods have been developed to enhance data analysis by optimizing the ordering of affinity matrices. Arabie and Carroll Arabie et al. (1978) introduced matrix permutation techniques to unveil block structures, thereby improving the understanding of network relationships. Wei et al. Wei et al. (2016) proposed GO-PQ, a strategy aimed at minimizing CPU cache misses by optimizing node arrangement. Zhao et al. Zhao et al. (2021) developed the DON model, which leverages a learned evaluation function to replace heuristics and capture the hidden locality of vertices. Further advancements include AutoLL by Watanabe et al. Watanabe & Suzuki (2021), which utilizes neural networks to reorder elements and elucidate the structures of adjacency matrices, and DeepTMR Watanabe & Suzuki (2022), which extracts nonlinear features for reordering based on a latent block model. However, these block diagonal generation methods generally require affinity matrix values to be limited to 0 or 1 and focus solely on rearranging the matrix without ensuring a fully block-diagonal form.

Block-diagonal identification, primarily developed for analyzing Hi-C data, has also seen significant advancements. Brault et al. Brault et al. (2017) explored methods for estimating block boundaries in diagonal blockwise matrices of Hi-C data, using a non-penalized approach to determine the number of block boundaries. They developed least squares estimators for both block boundaries and the number of blocks, which are theoretically proven to be consistent. Building on this work, Brault et al. Brault et al. (2018) introduced a nonparametric method for estimating block boundary locations in large Hi-C data matrices. However, these methods are specifically tailored for Hi-C data and lack broader applicability.

## 3 METHODOLOGY

In this section, we first introduce a method for permuting the affinity matrix into a block-diagonal form using density-based traversal. Following this, we outline the process for identifying diagonal blocks within the affinity matrix.

### 3.1 PERMUTING AFFINITY MATRIX INTO BLOCK-DIAGONAL FORM USING DENSITY-BASED TRAVERSAL

Consider a graph where the weights are defined by the affinity matrix; we begin with an affinity matrix-based density analysis. Subsequently, we introduce a density-based traversal algorithm designed to traverse all nodes within one cluster in the graph before proceeding to another. By reordering the row and column of the affinity matrix according to the traversal order, the affinity matrix can be permuted into a block-diagonal form.

### 3.1.1 AFFINITY MATRIX-BASED DENSITY ANALYSIS:

For a node $i$, the weight (or similarity) $w_{i,j}$ in the affinity matrix $\mathbf{W}$ indicates that node $j$ is closer to node $i$ when $w_{i,j}$ is larger. Let $c_i$ denote the weight between the $i$th node and its $\delta$th-largest-similarity neighbor, where $\delta$ is a positive integer parameter. Thus, a larger $c_i$ implies that node $i$ is located in a denser region. Clearly, points with lower density have smaller values of $c_i$, while points with higher density have larger values of $c_i$. This way, the density information of different clusters is encapsulated within the $c_i$ values of different nodes.

**Definition 1** (reachable similarity). *The reachable similarity of $j$th node to the $i$th node is defined as $s_{i,j} = \min(c_i, w_{i,j})$.*

Intuitively, for a node $i$, any node $j$ satisfying $w_{i,j} > c_i$ is considered equally as a close neighbor of node $i$, without any discrimination. However, for nodes $j$ satisfying $w_{i,j} < c_i$, the closeness to node $i$ is determined by the magnitude of $w_{i,j}$.

We propose a methodical approach to sequentially process clusters based on their density characteristics, i.e., reachable similarity and $\delta$th-largest-similarity. Our method begins by identifying a starting point within a high-density cluster, specifically selecting the point with the highest $c_i$. The process then involves an iterative exploration of neighboring points that exhibit a high degree of reachable similarity, continuing until no unprocessed points remain that are connected to already processed points. This exploration is conducted for each cluster until all points in the dataset have been evaluated.

By focusing on reachable similarity, our procedure systematically progresses through clusters, thus preventing the misclassification of distinct clusters as a single entity due to intervening noise points. The process initiates from the highest density node and proceeds to explore all connected points within the same cluster. Following the completion of one cluster, the method advances to an adjacent cluster, and repeats the process until all clusters are processed. This systematic exploration effectively manages varying densities and facilitates more precise cluster delineation. Importantly, our method does not require direct cluster assignment during the process, thus bypassing the output of conventional clustering results.

### 3.1.2 DENSITY-BASED TRAVERSAL ALGORITHM:

Based on the preceding analysis, the following traversal procedure has been developed. Initially, all points are designated as unprocessed. The cluster order expansion procedure is initiated by selecting an unprocessed core point with the highest density indicator $c_i$. This point is then marked as processed and appended to the order list $O$.

Subsequently, a priority queue $Q$ is instantiated and remains empty until the indices of all nodes within the $\delta$-neighborhood of the selected core point are enqueued. These nodes are ordered in descending sequence based on their existing reachable similarity $s_j$. The procedure continues as long as $Q$ is not empty, involving the subsequent steps: a) Dequeue the element $m$ from $Q$, which has the highest existing reachable similarity. b) For each index $j$ within the $\delta$-neighborhood of the element $m$, the existing similarity $s_j$ is updated if the reachable similarity between index $j$ and the node $m$ exceeds the current value of $s_j$. c) If the $m$-th node qualifies as a core point, then all indices of its $\delta$-neighborhood are re-enqueued into $Q$, where the queue maintains an automatic sort based on existing reachable similarity. Upon completion of these steps, the order of clusters is systematically documented in the list $O$. Consequently, the affinity matrix $\mathbf{W}$, when permuted according to the order list $O$, will manifest a block-diagonal structure.

The parameter $\delta$ is used to indicate the density of nodes in the region where each node is located. It does not require a precise setting. This paper proposes a rough yet effective approach for determining $\delta$. Specifically, we first calculate the average distance between nodes. We then use this value as the $c_i$ value for each point and calculate the recommended $\delta$ for each point. The final $\delta$ value is the average of all recommended $\delta$ values, i.e., $\delta = \frac{1}{N} \sum_{i=1}^{N} \arg\min_j \left( \left| w_{ij}^{dec} - \bar{w} \right| \right)$, where $\bar{w} = \frac{1}{N^2} \sum_{a=1}^{N} \sum_{b=1}^{N} w_{ab}$ is the average weight. We reorder $\{w_{i,1}, w_{i,2}, \ldots, w_{i,N}\}$ in descending order to obtain $\{w_{i,1}^{dec}, w_{i,2}^{dec}, \ldots, w_{i,N}^{dec}\}$.

The cluster ordering of a dataset can be graphically represented and interpreted. Let the traversal order be stored in $o$. The potential $L$ boundaries between diagonal blocks are then identified as the troughs in the curve $\{c_{o(i)} | i \in \{1, 2, \ldots, N\}\}$. It is important to note that these boundaries are not entirely precise, as they are based on local information and do not account for a global trade-off across all diagonal blocks.

While the traversal procedure bears certain similarities to DBSCAN in that both employ a density-based search within the graph, DBSCAN is reliant upon parameters such as eps and minPts. These parameters are often challenging to determine, and the clustering performance is highly sensitive to their configuration. Furthermore, DBSCAN may fail to distinguish between clusters separated by

regions of low density. By contrast, the proposed procedure obviates the need for manual parameter tuning, instead focusing solely on traversing data points to establish a traversal order without directly yielding clustering results.

## 3.2 IDENTIFYING DIAGONAL BLOCKS IN THE AFFINITY MATRIX

In this section, we aim to delineate the clustering outcome by pinpointing the diagonal blocks in the permuted affinity matrix. It is identified that only $K - 1$ partition indices $\{t_1, t_2, \ldots, t_{K-1}\} \subset \{1, 2, \ldots, N\}$ are required, which segregate the affinity matrix into a *block-diagonal* configuration with $K$ distinct blocks. These indices are ordered and unique, ensuring that each cluster contains at least one node and $t_{k-1} < t_k$ for all $k \in \{1, 2, \ldots, K\}$. Auxiliary indices are defined as $t_0 := 0$ and $t_K := N$ to facilitate analysis.

**1) Optimization Target:** The objective is to maximize the internal similarity of the diagonal blocks by determining the partition indices $\{t_k\}_{k=1}^{K-1}$. Let $\{\tau_k\}$ represent the segmentation variables, where the $k$-th diagonal block comprises data points indexed by $\mathcal{C}_k = \{\tau_{k-1} + 1, \ldots, \tau_k\}$. The sum of weights within the $k$-th block is given by $\sum_{i,j \in \mathcal{C}_k} w_{i,j}$. Our initial approach is to maximize the total weight across all blocks, expressed as $\sum_{k=1}^{K} \sum_{i,j \in \mathcal{C}_k} w_{i,j}$. To mitigate potential biases towards smaller clusters, a normalization term $\sum_{i \in \mathcal{C}_k} \sum_{j=1}^{N} w_{i,j}$ is incorporated into the objective function, ensuring a balanced consideration of cluster sizes. Mathematically, the problem is formulated as

$$\underset{\{\tau_k\}_{k=1}^{K-1}}{\text{maximize}} \quad \sum_{k=1}^{K} \frac{\sum_{i,j \in \mathcal{C}_k} w_{i,j}}{\sum_{i \in \mathcal{C}_k} \sum_{j=1}^{N} w_{i,j}}, \text{subject to} \quad 0 < \tau_1 < \tau_2 < \ldots < \tau_{K-1} < N \qquad (1)$$

Problem (1) can be reformulated as minimizing the normalized cut (Ncut) value across clusters, expressed as $\sum_{k=1}^{K} \frac{cut(\mathcal{C}_k, \bar{\mathcal{C}}_k)}{vol(\mathcal{C}_k)}$. Here, the term $cut(\mathcal{C}_k, \bar{\mathcal{C}}_k)$ is defined as $\sum_{i \in \mathcal{C}_k} \sum_{j \in \{1,2,\ldots,N\}, j \notin \mathcal{C}_k} w_{i,j}$, and $vol(\mathcal{C}_k)$ is given by $\sum_{i \in \mathcal{C}_k} \sum_{j \in \{1,2,\ldots,N\}} w_{i,j}$, aligning with the traditional Ncut problem as discussed in Shi & Malik (2000). Despite extensive studies over many years, the NP-hard characteristic of the Ncut problem limits solutions to approximations, typically via methods like spectral clustering. This paper proposes a novel approach whereby, under the assumption that $\mathbf{W}$ is block-diagonal, an optimal solution to the Ncut problem may be achieved in cases of well-separated clusters.

Define the block function as

$$f_k(\tau_k; \boldsymbol{\tau}_{-k}) \triangleq \frac{\sum_{i,j \in \mathcal{C}_k} w_{i,j}}{\sum_{i=\tau_{k-1}+1}^{\tau_k} \sum_{j=1}^{N} w_{i,j}} + \frac{\sum_{i,j \in \mathcal{C}_{k+1}} w_{i,j}}{\sum_{i=\tau_k+1}^{\tau_{k+1}} \sum_{j=1}^{N} w_{i,j}}$$

for $k = 1, 2, \ldots, K - 1$, where $\boldsymbol{\tau}_{-k} \triangleq (\tau_{k-1}, \tau_{k+1})$. In addition, define

$$f_0(\tau_0) = \frac{\sum_{i,j \in \{1,2,\ldots,\tau_1\}} w_{i,j}}{\sum_{i=1}^{\tau_1} \sum_{j=1}^{N} w_{i,j}}, f_K(\tau_K) = \frac{\sum_{i,j \in \{\tau_{K-1}+1, \tau_{K-1}+2, \ldots, N\}} w_{i,j}}{\sum_{i=\tau_{K-1}+1}^{N} \sum_{j=1}^{N} w_{i,j}}$$

for mathematical convenience. The problem (1) is equivalent to

$$\underset{\{\tau_k\}_{k=1}^{K-1}}{\text{maximize}} \quad \frac{1}{2} \sum_{k=0}^{K} f_k(\tau_k; \boldsymbol{\tau}_{-k}), \text{subject to} \quad 0 < \tau_1 < \tau_2 < \ldots < \tau_{K-1} < N$$

In the remaining part of the paper, we may omit the argument $\boldsymbol{\tau}_{-k}$ and write $f_k(\tau_k)$ for simplicity, as long as it is clear from the context.

**2) Properties of the $f_k(\tau_k)$ in cases of well-separated clusters.** Consider that the clusters are well-separated, resulting in the weights in $\mathbf{W}$ between clusters being zero. Suppose the clusters have been ordered correctly. The intra-cluster similarity for the $k$th cluster can be assumed to be $w_{i,j} = \mu_k$, where $i, j \in \{t_{k-1}+1, t_{k-1}+2, \ldots, t_k\}$, and $k \in \{1, 2, \ldots, K\}$, while the inter-cluster similarity is zero. Under these conditions, we observe the following properties for $f_k(\tau_k)$.

**Proposition 1** (Unimodality). *Suppose that, for some $k$, $\tau_{k-1}, \tau_{k+1} \in \{t_0, t_1, \ldots, t_K\}$, there exists only one index $t_j$, $j \in \{1, 2, \ldots, K-1\}$, within the interval $(\tau_{k-1}, \tau_{k+1})$. Then, $f_k(\tau) - f_k(\tau-1) > 0$ for $\tau_{k-1} < \tau < t_j$, and $f_k(\tau) - f_k(\tau - 1) < 0$ for $t_j < \tau < \tau_{k+1}$. In addition, $t_j$ minimizes $f_k(\tau)$ in $(\tau_{k-1}, \tau_{k+1})$. (Proof see Appendix A.1.)*

This result implies that, once the condition is satisfied, there exists an unique local minima $t_j$ of $f_k(\tau)$ over $(\tau_{k-1}, \tau_{k+1})$.

**Proposition 2** (Flatness). *Suppose that, for some $k$, $\tau_{k-1}, \tau_{k+1} \in \{t_0, t_1, ..., t_K\}$, there is no index $t_j \in \{t_1, t_2, \ldots, t_{K-1}\}$ in the interval $(\tau_{k-1}, \tau_{k+1})$. Then, $f_k(\tau) - f_k(\tau - 1)$ is constant for $\tau \in (\tau_{k-1}, \tau_{k+1})$. (Proof see Appendix 2.)*

It follows that, when the interval $(\tau_{k-1}, \tau_{k+1})$ does not contain $t_j$, the function $f_k(\tau)$ appears as a flat function for $\tau \in (\tau_{k-1}, \tau_{k+1})$.

**Proposition 3** (Monotonicity). *Suppose that, for some $k$, $\tau_{k-1}, \tau_{k+1} \in \{t_1, t_2, ..., t_{K-1}\}$, there are multiple partition indexes $t_j, t_{j+1}, \ldots, t_{j+J} \in \{t_0, t_1, \ldots, t_K\}$ within the interval $(\tau_{k-1}, \tau_{k+1})$. Then, $f(\tau) - f(\tau - 1) > 0$ for $\tau \in [\tau_{k-1}, t_j]$, and $f(\tau) - f(\tau - 1) < 0$ for $\tau \in [t_{j+J}, \tau_{k+1}]$. Moreover, for any interval $(t_k, t_{k+1})$, $k \in \{j, j, ..., j + J - 1\}$, there exists a constant $\hat{\tau}$, such that*

*1) If $\hat{\tau} \in (t_k, t_{k+1})$, then $f(\tau) - f(\tau - 1) > 0$ for $\tau \in [t_k, \hat{\tau}]$ and $f(\tau) - f(\tau - 1) < 0$ for $\tau \in [\hat{\tau}, t_{k+1}]$;*

*2) If $\hat{\tau} \in [t_{k+1}, N)$, then $f(\tau) - f(\tau - 1) > 0$ for $\tau \in [t_k, t_{k+1}]$;*

*3) If $\hat{\tau} \in (0, t_k]$, then $f(\tau) - f(\tau - 1) < 0$ for $\tau \in [t_k, t_{k+1}]$. (Proof see Appendix A.3.)*

**3) Split-and-Refine Algorithm.** Based on the above property, we develop a method to address problem (1), providing an optimal solution in cases of well-separated clusters. The method iteratively alternates between introducing a new segmentation to the existing set and refining the positions of all segmentations until a stationary state is reached, that is, until no further modifications to any segmentation result in an improvement in the objective function value.

In the $m$-th iteration, where $m = 1, 2, 3, \ldots$, there exist $m$ intervals $(\tau_{k-1}, \tau_k)$ for $k \in \{1, 2, \ldots, m\}$ with $\tau_m = N$. The $k$-th interval among these $m$ intervals is selected for division into two new intervals, thereby expanding the set to $m + 1$ intervals. The segmentation indices corresponding to this configuration are represented by an $m$-tuple $\boldsymbol{\tau}^{(m,k)} = (\tau_1^{(m,k)}, \tau_2^{(m,k)}, \ldots, \tau_m^{(m,k)})$. The function $f_k(\tau; \boldsymbol{\tau}_{-k}^{(m,k)})$ is then maximized subject to the constraint $\tau \in (\tau_{k-1}^{(m,k)}, \tau_{k+1}^{(m,k)})$, with the minimal value being denoted as $f_*^{(m,k)}$. In addition, denote the benefit of splitting as

$$\triangle f_*^{(m,k)} = f_*^{(m,k)} - f_k(\tau_{k-1}^{(m,k)}; \boldsymbol{\tau}_{-k}^{(m,k)}) \tag{2}$$

where $f_k(\tau_{k-1}^{(m,k)}; \boldsymbol{\tau}_{-k}^{(m,k)})$ represents the objective function value in the absence of any split.

Consequently, $\Delta f_*^{(m,k)}$ quantifies the incremental benefit derived from optimally splitting the $k$-th interval among the $m$ intervals. To ascertain the most advantageous segmentation, the benefit increase is evaluated across all possible $m$ combinations of the split. This iterative evaluation facilitates the identification of the optimal segmentation variable $\boldsymbol{\tau}^{(m+1)}$, which maximizes the overall benefit. This entire procedure is methodically outlined in Alg. 1 in the appendix.

The proposed splitting procedure operates by sequentially searching for segmentation adjustments. However, it cannot be guaranteed that the resultant set of segmentations is stationary. To address this, we introduce a refining process post-insertion of each new segmentation. This refining stage entails iterating over the current segmentations, individually refining each to maximize the objective function. The objective may either increase or remain unchanged during this process, and refinement continues until no further changes in the segmentation can be made. Notably, if no modifications have occurred in the segmentations, there is no necessity for multiple refining calls.

In practical applications, the exact number of clusters is often undetermined. Therefore, we handle scenarios where the number of clusters, $K$, is unknown by setting an upper limit, $L$. Segmentations are inserted sequentially until the number of segmentations reaches $L$. Within Alg. 1, lines 6 and 7 can execute in parallel across the $m$ segments. Additionally, the refinement of the $m - 1$ segmentations can be parallelized by alternating between refining even and odd-numbered segmentations until a stationary state is achieved.

To determine the optimal number of clusters, we record the maximum objective value for problem (1) as $g(m)$ for cluster numbers ranging from $m = 1$ to $m = L - 1$, under the assumption that the objective value escalates with an increase in $K$. The identification of the *inflection point* on

the curve $g(m)$ is facilitated through the computation of the second derivative $g_m^{''} = (g(m) - g(m-1)) - (g(m+1) - g(m))$. The optimal number of clusters is then determined as $K^* = \arg\max_{K \in \{2,3,...,L-1\}} g_m^{''}$.

**Lemma 1** (Cost Reduction). *Consider two distinct intervals $(\tau_{k-1}^{(m,k)}, \tau_{k+1}^{(m,k)})$ and $(\tau_{k'-1}^{(m,k')}, \tau_{k'+1}^{(m,k')})$ constructed from the mth iteration of Step 1) in Alg. 1 in the appendix, where $\tau_{k-1}^{(m,k)}, \tau_{k+1}^{(m,k)}, \tau_{k'-1}^{(m,k')}, \tau_{k'+1}^{(m,k')} \in t_0, t_1, ..., t_K$. Suppose that there exists at least one index $t_j \in \{t_1, t_2, \ldots, t_{K-1}\}$ in $(\tau_{k-1}^{(m,k)}, \tau_{k+1}^{(m,k)})$, and no such $t_j$ in $(\tau_{k'-1}^{(m,k')}, \tau_{k'+1}^{(m,k')})$. Then, $\triangle f_*^{(m,k)} > \triangle f_*^{(m,k')}$. (Proof see Appendix A.4.)*

Lemma 1 can be intuitively understood from Propositions 1 and 2, which suggest that $f_k(\tau; \boldsymbol{\tau}_{-k}^{(m,k)})$ is unimodal in $(\tau_{k-1}^{(m,k)}, \tau_{k+1}^{(m,k)})$, but $f_{k'}(\tau; \boldsymbol{\tau}_{-k'}^{(m,k')})$ is flat in $(\tau_{k'-1}^{(m,k')}, \tau_{k'+1}^{(m,k')})$, and hence, the former one has a larger potential to increase the total benefit $\sum_{k=0}^{m} f_k(\tau_k; \boldsymbol{\tau}_{-k})$.

**Theorem 4** (Optimality). *The proposed split-and-refine Alg. 1 in the appendix will output $\boldsymbol{\tau}^* = (\tau_1^*, \tau_2^*, \ldots, \tau_{K-1}^*)$, with $\tau_k^* = t_k$, $k = 1, 2, \ldots, K-1$. (Proof see Appendix A.5.)*

Theorem 4 provides the optimality guarantee of the proposed split-and-merge method in cases of well-separated clusters. Finally, the clustering assignment is given by the segmentation and the transversal order.

## 4 EXPERIMENTAL RESULT

### 4.1 EXPERIMENTAL SETUP

#### 4.1.1 DATASET

We extensively evaluated the proposed DAM algorithm on six publicly available datasets: **CIFAR-100** Krizhevsky et al. (2009), consisting of 60,000 images of 100 objects, each of size $32 \times 32$ pixels, categorized into 100 classes; **ImageNet-10** Chang et al. (2017), which includes 13,000 images of 10 objects selected from the ILSVRC2012 1K dataset Deng et al. (2009), each with dimensions of $224 \times 224$ pixels; **EYaleB** Georghiades et al. (2001), comprising 2,432 images of 38 subjects under 9 illumination conditions, downsampled to $48 \times 42$ pixels following Ji et al. (2017b); **MNIST** LeCun et al. (1998), which contains 70,000 grayscale images, each $28 \times 28$ pixels, categorized into 10 classes, and preprocessed using scattered convolutional features Bruna & Mallat (2013) with PCA for dimensionality reduction to 2000; **COIL-100** Nene et al. (1996), which has 7,200 images of 100 objects, each of size $128 \times 128$ pixels, taken at 5-degree pose intervals; and **ORL** Samaria & Harter (1994), consisting of 400 face images of 40 subjects, each of size $112 \times 92$ pixels, with variations in expressions, lighting, and accessories.

#### 4.1.2 COMPARISONS

We compare with fifty-three existing state-of-the-art approaches including: S$^5$C Matsushima & Brbic (2019), SSCOMP You et al. (2016b), SC-LALRG Yin et al. (2018) , KCRSC Wang et al. (2018), S$^3$COMP-C Chen et al. (2020) , FTRR Ma et al. (2020) , PSSC$_l$ Lv et al. (2022), PSSC Lv et al. (2022), DCFSC Seo et al. (2019), Struct-AE Peng et al. (2018) , DEC Xie et al. (2016), IDEC Guo et al. (2017), SR-SSC Abdolali et al. (2019), EDESC Cai et al. (2023), EnSC-ORGEN You et al. (2016a), NCSC Zhang et al. (2019c), DSC-Net-L1 Ji et al. (2017a), ACC_CN Li et al. (2020b), DSC-Net-L2 Ji et al. (2017a), DLRSC Kheirandishfard et al. (2020a), RGRL-L2 Kang et al. (2020), ODSC Valanarasu & Patel (2021), MESC-NetPeng et al. (2022), Cluster-GAN Ghasedi et al. (2019), DEPICT Ghasedi Dizaji et al. (2017), SENet Zhang et al. (2022), SpecNet Shaham et al. (2018), S$^2$Conv-SCN-L2 Zhang et al. (2019b), S$^2$Conv-SCN-L1 Zhang et al. (2019b), RED-SC Yang et al. (2020), DASC Zhou et al. (2018), MLRDSC Kheirandishfard et al. (2020b), DSC-DLHuang et al. (2020), MLRDSC-DA Abavisani et al. (2020), DAE Vincent et al. (2010), DCGAN Radford et al. (2015), DeCNN Zeiler et al. (2010), JULE Yang et al. (2016), VAE Kingma & Welling (2013), ADC Haeusser et al. (2019), AE Bengio et al. (2006), DAC Chang et al. (2017), IIC Ji et al. (2019), DCCM Wu et al. (2019), PICA Huang et al. (2020), CC Li et al. (2021), SPICE Niu et al. (2023), SCAN Van Gansbeke et al. (2020), PCL Li et al. (2020a),

TCL Li et al. (2022), RCFE Li et al. (2018), S²ESC Zhu et al. (2021), SSRSC Xu et al. (2019).

In all experiments, clustering accuracy (Acc) and normalized mutual information (NMI) are employed as evaluation metrics. The performance data for the baseline methods is sourced from their original publications. Notably, the proposed DAM method operates without the need for manually set parameters. The affinity matrix employed in *DAM* is constructed using BDR-B Lu et al. (2018), a classical and effective method that incorporates block-diagonal priors. The primary focus of this paper does not lie in the construction techniques for the affinity matrix. The results in Tab. 1 and Tab. 2 will demonstrate that the proposed *DAM* yields substantial performance improvements compared to the use of BDR-B solely.

Table 1: Comparison of the proposed DAM algorithm with existing SOTA methods across various datasets.

| Methods | EYaleB ACC | EYaleB NMI | MNIST ACC | MNIST NMI | ORL ACC | ORL NMI | COIL-100 ACC | COIL-100 NMI |
|---|---|---|---|---|---|---|---|---|
| S⁵C | 60.70 | - | 59.60 | - | - | - | 54.10 | - |
| SSCOMP | 77.59 | 83.25 | - | - | - | - | - | - |
| SC-LALRG | 79.66 | 84.52 | 78.20 | 76.01 | - | - | - | - |
| KCRSC | 81.40 | 88.10 | 64.70 | 64.30 | 72.30 | 86.30 | - | - |
| S³COMP-C | 87.41 | 86.32 | 96.32 | - | - | - | 78.89 | - |
| FTRR | - | - | 70.70 | 66.72 | - | - | - | - |
| PSSC$_l$ | - | - | 78.50 | 72.76 | 85.25 | 92.58 | - | - |
| PSSC | - | - | 84.30 | 76.76 | 86.75 | 93.49 | - | - |
| DCFSC | 93.87 | - | - | - | 85.20 | - | 72.70 | - |
| Struct-AE | 94.70 | - | 65.70 | 68.98 | - | - | - | - |
| IDEC | - | - | 88.06 | 86.72 | - | - | - | - |
| SR-SSC | - | - | 91.09 | **93.06** | - | - | - | - |
| EDESC | - | - | 91.30 | 86.20 | - | - | - | - |
| EnSC-ORGEN | - | - | 93.79 | - | - | - | 69.24 | - |
| NCSC | - | - | 94.09 | 86.12 | - | - | - | - |
| DSC-Net-L2 | 97.33 | - | - | - | 86.00 | - | 69.04 | - |
| ACC_CN | 97.31 | 99.34 | 78.60 | 74.21 | - | - | - | - |
| DLRSC | 97.53 | - | - | - | - | - | 71.86 | - |
| RGRL-L2 | 97.53 | 96.61 | 81.40 | 75.52 | - | - | - | - |
| ODSC | 97.78 | - | 81.20 | - | - | - | - | - |
| MESC-Net | 98.03 | 97.27 | 81.11 | 82.26 | - | - | 71.88 | 90.76 |
| Cluster-GAN | - | - | 96.40 | 92.10 | - | - | - | - |
| DEPICT | - | - | 96.50 | 91.70 | - | - | - | - |
| BDR-B | 82.51 | 79.15 | 67.55 | 72.83 | 70.54 | 74.26 | 71.56 | 82.11 |
| SENet | - | - | 96.80 | 91.80 | - | - | - | - |
| SpecNet | - | - | 97.10 | 92.40 | - | - | - | - |
| S²Conv-SCN-L1 | 98.48 | - | - | - | 89.50 | - | 73.33 | - |
| RED-SC | 98.52 | - | 74.34 | 73.16 | 86.13 | 91.16 | - | - |
| DASC | 98.56 | 98.01 | 80.40 | 78.00 | 88.25 | 93.15 | - | - |
| MLRDSC | 98.64 | - | - | - | 88.75 | - | 76.72 | - |
| DSC-DL | 98.90 | 97.40 | 81.20 | 76.10 | - | - | - | - |
| MLRDSC-DA | 99.18 | - | - | - | - | - | 79.33 | - |
| RCFE | - | - | - | - | - | - | 79.63 | **96.23** |
| S²ESC | - | - | - | - | 89.00 | 93.52 | - | - |
| SSRSC | - | - | - | - | 78.25 | - | - | - |
| DAM | **99.95** | **99.95** | **97.35** | 92.81 | **90.75** | **94.66** | **84.95** | 93.91 |

## 4.2 EVALUATIONS ON DIFFERENT DATASETS

**EYaleB dataset:** As shown in Tab. 1, the proposed DAM achieves an accuracy of 99.95% and an NMI of 99.95% on the EYaleB dataset, surpassing all baseline methods. MLRDSC-DA Abavisani et al. (2020) records the second-highest accuracy at 99.18%, while ACC_CN Li et al. (2020b) achieves the second-best NMI performance at 99.34%. The DAM method outperforms the second-best by 0.77% in accuracy and 0.61% in NMI.

**MNIST dataset:** As indicated in Tab. 1, DAM attains an accuracy of 97.35% and an NMI of 92.81% on the MNIST dataset. SpecNet Shaham et al. (2018) achieves the second-best accuracy at 97.10%, and SR-SSC Abdolali et al. (2019) achieves the highest NMI at 93.06%. DAM slightly exceeds SpecNet in accuracy by 0.25% but trails SR-SSC in NMI by 0.25%.

**ORL dataset:** The proposed DAM achieves 90.75% accuracy and 94.66% NMI on the ORL dataset, outperforming all baselines as shown in Tab. 1. Specifically, DAM surpasses the second-best, S²Conv-SCN-L2 Zhang et al. (2019b), by 1.25% in accuracy, and S²ESC Zhu et al. (2021) by 1.14% in NMI.

Table 2: Comparison of the proposed DAM algorithm with existing SOTA methods across various datasets.

| Methods | CIFAR-100 ACC | CIFAR-100 NMI | ImageNet-10 ACC | ImageNet-10 NMI |
|---|---|---|---|---|
| DEC | 18.5 | 13.6 | 38.1 | 28.2 |
| DAE | 15.1 | 11.1 | 30.4 | 20.6 |
| DCGAN | 15.1 | 12.0 | 34.6 | 22.5 |
| DeCNN | 13.3 | 9.2 | 31.3 | 18.6 |
| JULE | 13.7 | 10.3 | 30.0 | 17.5 |
| VAE | 15.2 | 10.8 | 33.4 | 19.3 |
| ADC | 16.0 | - | - | - |
| AE | 16.5 | 10.00 | 31.7 | 21.0 |
| DAC | 23.8 | 18.5 | 52.7 | 39.4 |
| BDR-B | 22.5 | 23.7 | 31.6 | 50.9 |
| IIC | 25.7 | - | - | - |
| DCCM | 32.7 | 28.5 | 71.0 | 60.8 |
| PICA | 33.7 | 31.0 | 87.0 | 80.2 |
| CC | 42.9 | 43.1 | 89.3 | 85.9 |
| SPICE | 46.8 | 44.8 | - | - |
| TCL | **53.1** | **52.9** | 89.5 | 87.5 |
| DAM | 47.75 | 45.77 | **91.69** | **87.53** |

**COIL-100 dataset:** For the COIL-100 dataset, as shown in Tab. 1, DAM achieves an accuracy of 84.95% and an NMI of 93.91%. While DAM leads in accuracy, RCFE Li et al. (2018) records the highest NMI at 96.23%, with DAM being the second-best in NMI.

**CIFAR-100 dataset:** As presented in Tab. 2, DAM achieves 47.75% accuracy and 45.77% NMI on the CIFAR-100 dataset, falling short by 5.35% in accuracy and 7.13% in NMI compared to the baseline TCL Li et al. (2022). Despite this, DAM outperforms all baselines except SCAN, with TCL's superior performance attributed to fine-tuned contrastive clustering.

**ImageNet-10 dataset:** DAM achieves 91.69% accuracy and 87.53% NMI on the ImageNet-10 dataset, as shown in Tab. 2, surpassing all state-of-the-art methods. It improves upon the second-best method, TCL Li et al. (2022), by 2.19% in accuracy and 0.03% in NMI.

### 4.3 Quantitative Result

Fig. 2 illustrates the block-diagonal form generated by the proposed density-based traversal algorithm, along with the block diagonal results produced by the split-and-refine algorithm. First, we observe that our method effectively orders the affinity matrix into a block-diagonal structure. Secondly, our approach demonstrates a high accuracy in segmenting the individual diagonal blocks.

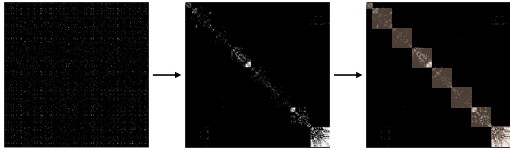

Figure 2: An example of the block-diagonal generation and segmentation procedure applied to the MNIST LeCun et al. (1998) dataset. White points indicate high similarity values, whereas black points represent zero similarity.

### 4.4 Ablation Study

Behold the results presented in Tab. 3, which elucidate the subtleties of block-diagonal identification and segmentation within the proposed *DAM*

Table 3: Ablation (ACC performance) of permutation (Perm.) and segmentation (Seg.) in the proposed DAM.

|              | CIFAR-100 | ImageNet-10 | EYaleB | MNIST | COIL-100 | ORL   |
|--------------|-----------|-------------|--------|-------|----------|-------|
| GO+Seg.      | 41.24     | 80.42       | 85.54  | 87.32 | 70.53    | 74.47 |
| DON-RL+Seg.  | 42.51     | 80.14       | 84.67  | 88.24 | 72.64    | 78.28 |
| DeepTMR+Seg. | 40.47     | 81.01       | 89.11  | 86.75 | 73.27    | 80.71 |
| Perm.+DBM    | 42.13     | 86.07       | 92.22  | 92.37 | 72.32    | 82.44 |
| Perm.+NMC    | 43.15     | 85.33       | 94.04  | 91.56 | 74.41    | 81.47 |
| DAM          | 47.75     | 91.69       | 99.95  | 97.35 | 90.75    | 84.95 |

algorithm. Although we are the pioneers in employing these techniques for clustering, we undertake experiments by substituting certain stages of the proposed method with several related existing graph ordering and segmentation techniques. When graph ordering methods such as GO Wei et al. (2016), DON-RL Zhao et al. (2021), and DeepTMR Watanabe & Suzuki (2022) are applied to the affinity matrix, followed by the proposed block-diagonal segmentation, a significant decline in performance is observed. This degradation arises because these methods were originally designed for value ordering rather than clustering. Likewise, employing DBM Brault et al. (2017) and NMC Brault et al. (2018) on the permuted affinity matrix also results in a marked reduction in performance, as these approaches were specifically developed for the unique structure of Hi-C matrices and do not accommodate the particular requirements of clustering tasks.

## 5 Conclusion

In this paper, we introduce a novel clustering method, termed *DAM*. This approach employs a cluster traversal algorithm to determine a permutation that reorders the affinity matrix into a *block-diagonal* structure. Subsequently, we propose a split-and-refine algorithm to identify the diagonal blocks within the permuted affinity matrix, with the clustering results derived from the successful identification of these blocks. The proposed *DAM* method consistently achieves the highest or second-best clustering performance across six real-world benchmark image clustering datasets, in comparison with state-of-the-art methods.

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

# A APPENDIX

## A.1 PROOF OF PROPOSITION 1

Without loss of generality, suppose $k = 1$, and $\tau_{k-1} = t_0, \tau_{k+1} = t_2, t_j = t_1$. Consider $\tau \in (0, t_1]$, we have

$$
\begin{aligned}
f_1(\tau) &= \frac{(t_1 - \tau)^2 \mu_1 + (t_2 - t_1)^2 \mu_2}{(t_1 - \tau)t_1 \mu_1 + (t_2 - t_1)^2 \mu_2} + \frac{t^2 \mu_1}{\tau t_1 \mu_1} \\
&= \frac{\tau}{t_1} + \frac{1 + (t_1 - \tau)^2 C_1}{1 + (t_1 - \tau)t_1 C_1}
\end{aligned}
$$

where $C_1 = \frac{\mu_1}{(t_2 - t_1)^2 \mu_2}$. Then, we have

$$
\begin{aligned}
f_1(\tau) &= \frac{\tau(1 + (t_1 - \tau)t_1 C_1) + t_1(1 + (t_1 - \tau)^2 C_1)}{t_1(1 + (t_1 - \tau)t_1 C_1)} \\
&= 1 + \frac{1}{(1 + t_1^2 C_1)t_1 \tau^{-1} - t_1^2 C_1}
\end{aligned}
$$

The function $f_1(\tau)$ is monotonically increasing for $\tau \in (0, t_1]$.

Consider $\tau \in [t_1, t_2)$, we have

$$
\begin{aligned}
f_1(\tau) &= \frac{(t_2 - \tau)^2 \mu_1}{(t_2 - \tau)(t_2 - t_1)\mu_1} \\
&\quad + \frac{t_1^2 \mu_1 + (t_2 - t_1)^2 \mu_2 + (\tau - t_1)^2 \mu_2}{t_1^2 \mu_1 + (t_2 - t_1)^2 \mu_2 + (\tau - t_1)(t_2 - t_1)\mu_2} \\
&= \frac{t_2 - \tau}{t_2 - t_1} + \frac{1 + (\tau - t_1)^2 C_2}{1 + (\tau - t_1)(t_2 - t_1)C_2}
\end{aligned}
$$

where $C_2 = \frac{\mu_2}{t_1^2 \mu_1 + (t_2 - t_1)^2 \mu_2}$. Then, we have

$$f_1(\tau) = 1 + \frac{t_2 - \tau}{t_2 - t_1 + (t_2 - t_1)^2 C_2 (\tau - t_1)}$$

$$= 1 + \frac{1}{\triangle_{21}(1 + \triangle_{21}^2 C_2)(t_2 - \tau)^{-1} - \triangle_{21}^2 C_2}$$

where $\triangle_{21} = t_2 - t_1$. The function $f_1(\tau)$ is monotonically decreasing for $\tau \in [t_1, t_2)$.

## A.2 PROOF OF PROPOSITION 2

Without loss of generality, suppose $k = 1$, and $\tau_{k-1} = t_0$, $\tau_{k+1} = t_1$.

$$f_1(\tau) = \frac{\sum_{i,j \in [1,\tau]} w_{i,j}}{\sum_{i \in [1,\tau]} \sum_{j \in [1,t_1]} w_{i,j}} + \frac{\sum_{i,j \in [\tau+1,t_1]} w_{i,j}}{\sum_{i \in [\tau+1,t_1]} \sum_{j \in [1,t_1]} w_{i,j}}$$

$$= \frac{\tau^2 \mu_1}{\tau t_1 \mu_1} + \frac{(t_1 - \tau)^2 \mu_1}{(t_1 - \tau) t_1 \mu_1} = 1$$

The function $f_1(\tau)$ is constant for $\tau \in (t_0, t_1)$.

## A.3 PROOF OF PROPOSITION 3

Without loss of generality, suppose $k = 1$, and $\tau_{k-1} = 0$, $\tau_{k+1} = t_L$, $L \geq 3$.

• Consider $\tau \in (0, t_1]$, we have

$$f_1(\tau) = \frac{(t_1 - \tau)^2 \mu_1 + (t_2 - t_1)^2 \mu_2 + ... + (t_L - t_{L-1})^2 \mu_L}{(t_1 - \tau) t_1 \mu_1 + (t_2 - t_1)^2 \mu_2 + ... + (t_L - t_{L-1})^2 \mu_L}$$

$$+ \frac{\tau^2 \mu_1}{\tau t_1 \mu_1}$$

$$= \frac{\tau}{t_1} + \frac{1 + (t_1 - \tau)^2 C_1}{1 + (t_1 - \tau) t_1 C_1}$$

where $C_1 = \frac{\mu_1}{(t_L - t_1)^2 b(2, L)}$ and $b(2, L) = \frac{(t_2 - t_1)^2 \mu_2 + ... + (t_L - t_{L-1})^2 \mu_L}{(t_L - t_1)^2}$. $f_1(\tau)$ can also be written as

$$f_1(\tau) = \frac{\tau(1 + (t_1 - \tau) t_1 C_1) + t_1(1 + (t_1 - \tau)^2 C_1)}{t_1(1 + (t_1 - \tau) t_1 C_1)}$$

$$= \frac{\tau + t_1 + (t_1 - \tau) t_1 C_1 (\tau + t_1 - \tau)}{t_1 + (t_1 - \tau) t_1^2 C_1}$$

$$= \frac{\tau + t_1 + (t_1 - \tau) t_1^2 C_1}{t_1 + (t_1 - \tau) t_1^2 C_1}$$

$$= 1 + \frac{1}{(1 + t_1^2 C_1) t_1 \tau^{-1} - t_1^2 C_1}$$

The function $f_1(\tau)$ is monotonically increasing for $\tau \in (0, t_1]$.

• Consider $\tau \in [t_{L-1}, t_L)$, we have

$$f_1(\tau) = \frac{(t_L - \tau)^2 \mu_1}{(t_L - \tau)(t_L - t_{L-1}) \mu_1} + \phi$$

$$= \frac{(t_L - \tau)^2 \mu_1}{(t_L - \tau)(t_L - t_{L-1}) \mu_1} + \phi$$

$$= \frac{t_L - \tau}{t_L - t_{L-1}} + \frac{1 + (\tau - t_{L-1})^2 C_2}{1 + (\tau - t_{L-1})(t_L - t_{L-1}) C_2}$$

where

$$C_2 = \frac{\mu_L}{t_{L-1}^2 b(1, L-1)},$$

$$\phi = \frac{t_{L-1}^2 b(1, L-1) + (\tau - t_{L-1})^2 \mu_L}{t_{L-1}^2 b(1, L-1) + (\tau - t_{L-1})(t_L - t_{L-1})\mu_L},$$

and

$$b(1, L-1) = \frac{t_1^2 \mu_1 + (t_2 - t_1)^2 \mu_2 + ... + (t_{L-1} - t_{L-2})^2 \mu_{L-1}}{t_{L-1}^2}.$$

$f_1(\tau)$ can also be written as

$$f_1(\tau) = 1 + \frac{t_L - \tau}{t_L - t_{L-1} + (t_L - t_{L-1})^2 C_2 (\tau - t_{L-1})}$$

$$= 1 + \frac{1}{\triangle_L (1 + \triangle_L^2 C_2)(t_L - \tau)^{-1} - \triangle_L^2 C_2}$$

where $\triangle_L = t_L - t_{L-1}$. The function $f_1(\tau)$ is monotonically decreasing for $\tau \in [t_{L-1}, t_L)$.

• Consider $\tau \in [t_k, t_{k+1}]$, for any $k \in \{1, 2, ..., L-2\}$.

$$f_1(\tau)$$

$$= \frac{t_1^2 \mu_1 + ... + (t_k - t_{k-1})^2 \mu_k + (\tau - t_k)^2 \mu_{k+1}}{t_1^2 \mu_1 + ... + (t_k - t_{k-1})^2 \mu_k + (t_{k+1} - t_k)(\tau - t_k)\mu_{k+1}}$$

$$+ \frac{(t_{l+2} - t_{k+1})^2 \mu_{l+2} + ...}{(t_{l+2} - t_{k+1})^2 \mu_{l+2} + ...}$$

$$= \frac{t_k^2 b(1, k) + (\tau - t_k)^2 \mu_{k+1}}{t_k^2 b(1, k) + (t_{k+1} - t_k)(\tau - t_k)\mu_{k+1}}$$

$$+ \frac{(t_L - t_{k+1})^2 b(k+2, L) + (t_{k+1} - \tau)^2 \mu_{k+1}}{(t_L - t_{k+1})^2 b(k+2, L) + (t_{k+1} - t)(t_{k+1} - t_k)\mu_{k+1}}$$

where $b(1, k) = \frac{t_1^2 \mu_1 + ... + (t_k - t_{k-1})^2 \mu_k}{t_k^2}$ and $b(k+2, L) = \frac{(t_{k+2} - t_{k+1})^2 \mu_{k+2} + ... + (t_L - t_{k-1})^2 \mu_L}{(t_L - t_{k+1})^2}$. Denote $d = t_{k+1} - t_k$, and $x = \tau - \frac{t_{k+1} + t_k}{2}$. Since $\tau \in (t_{k+1}, t_k)$, we have $x \in (-\frac{d}{2}, \frac{d}{2})$. Denote $B_1 = \frac{\mu_{k+1}}{t_k^2 b(1,k)}$, and $B_2 = \frac{\mu_{k+1}}{(t_L - t_{k+1})^2 b(k+2,L)}$. So, $f_1(\tau)$ can be written as

$$f(x) = \frac{x^2(-d^2 B_1 B_2 + B_1 + B_2) + 2dx(B_1 - B_2) + \varphi_1}{-x^2 d^2 B_1 B_2 + xd(B_1 - B_2) + \varphi_2}$$

where $\varphi_1 = \frac{3}{4}d^2(B_1 + B_2) + \frac{d^4}{4}B_1 B_2 + 2$, $\varphi_2 = \frac{d^4}{4}B_1 B_2 + \frac{1}{2}d^2(B_1 + B_2) + 1$. $f(x)$ is monotonically decreasing for $x \leq x_0$, and monotonically increasing for $x \geq x_0$, where

$$x_0 = \begin{cases} \frac{(\sqrt{B_1 d^2 + 1} + \sqrt{B_2 d^2 + 1})^2}{2d(B_2 - B_1)} & B_1 \neq B_2 \\ 0 & B_1 = B_2 \end{cases}$$

Recall the definition of $d$ and $x$, the function $f_1(\tau)$ is monotonically decreasing for $\tau \leq \hat{\tau}$, and monotonically increasing for $\tau \geq \hat{\tau}$, where

$$\hat{\tau} = \frac{\left(\sqrt{B_1 (t_{k+1} - t_k)^2 + 1} + \sqrt{B_2 (t_{k+1} - t_k)^2 + 1}\right)^2}{2(t_{k+1} - t_k)(B_2 - B_1)}$$

$$+ \frac{1}{2}(t_{k+1} + t_k)$$

for $B_1 \neq B_2$, and $\frac{1}{2}(t_{k+1} + t_k)$ otherwise. Thus, for the interval $[t_k, t_{k+1}]$, $k = 1, ..., L-2$, $f_1(\tau)$ increases for $\tau \in [t_k, \hat{\tau}]$ and decreases for $\tau \in [\hat{\tau}, t_{k+1}]$ if $\hat{\tau} \in (t_k, t_{k+1})$; $f_1(\tau)$ increases for $\tau \in [t_k, t_{k+1}]$ if $\hat{\tau} \geq t_{k+1}$; $f_1(\tau)$ decreases for $\tau \in [t_k, t_{k+1}]$ if $\hat{\tau} \leq t_k$.

Consider two distinct intervals $(\tau_{k-1}^{(m,k)}, \tau_{k+1}^{(m,k)})$ and $(\tau_{k'-1}^{(m,k')}, \tau_{k'+1}^{(m,k')})$ constructed from the $m$th iteration of Step 1) in Alg. 1, where $\tau_{k-1}^{(m,k)}, \tau_{k+1}^{(m,k)}, \tau_{k'-1}^{(m,k')}, \tau_{k'+1}^{(m,k')} \in t_0, t_1, ..., t_K$. Suppose that there exists at least one index $t_j \in \{t_1, t_2, \ldots, t_{K-1}\}$ in $(\tau_{k-1}^{(m,k)}, \tau_{k+1}^{(m,k)})$, and no such $t_j$ in $(\tau_{k'-1}^{(m,k')}, \tau_{k'+1}^{(m,k')})$. Then, $\triangle f_*^{(m,k)} > \triangle f_*^{(m,k')}$.

## A.4 Proof of Lemma 1

Since there exists at least one index $t_j \in \{t_1, t_2, \ldots, t_{K-1}\}$ in $(\tau_{k-1}^{(m,k)}, \tau_{k+1}^{(m,k)})$, it thus follows from Proposition 3 that $\triangle f_*^{(m,k)} = f_*^{(m,k)} - f_k(\tau_{k-1}^{(m,k)}; \boldsymbol{\tau}_{-k}^{(m,k)}) > f_k(\tau; \boldsymbol{\tau}_{-k}^{(m,k)}) - f_k(\tau_{k-1}^{(m,k)}; \boldsymbol{\tau}_{-k}^{(m,k)})$.

Since there exists no such $t_j$ in $(\tau_{k'-1}^{(m,k')}, \tau_{k'+1}^{(m,k')})$, it thus follows from Proposition 2 that $\triangle f_*^{(m,k')} = f_*^{(m,k')} - f_{k'}(\tau_{k'-1}^{(m,k')}; \boldsymbol{\tau}_{-k'}^{(m,k')}) = f_{k'}(\tau; \boldsymbol{\tau}_{-k'}^{(m,k')}) - f_{k'}(\tau_{k'-1}^{(m,k')}; \boldsymbol{\tau}_{-k'}^{(m,k')})$.

As a result, $\triangle f_*^{(m,k)} > \triangle f_*^{(m,k')}$.

## A.5 Proof of Theorem 4

For $m = 1$, we split the interval $(t_0, t_K)$ into two subintervals. Proposition 1 indicates that one of $\{t_1, t_2, ..., t_{K-1}\}$ will be the first optimal split index.

For $m = 2$, suppose the first optimal split index is $t_1$, and $\boldsymbol{\tau}^{(2)} = \{t_1\}$. We then insert the second split index into the intervals $(t_0, t_1)$ and $(t_1, t_K)$. Lemma 1 indicates that the larger $\triangle f_*^{(2,k)}$ arises from splitting the interval $(t_1, t_K)$, which contains at least one of $\{t_2, t_3, ..., t_{K-1}\}$.

For $m = 3$, suppose the second optimal split index is $t_2$, and $\boldsymbol{\tau}^{(3)} = \{t_1, t_2\}$. We then insert the third split index into the intervals $(t_0, t_1)$, $(t_1, t_2)$, and $(t_2, t_K)$. The larger $\triangle f_*^{(3,k)}$ arises from splitting the interval that contains at least one of $\{t_3, t_4, ..., t_{K-1}\}$.

We repeat this process until completing iteration $m = K - 1$. Then, we have $\boldsymbol{\tau}^{(K)} = \{t_1, t_2, ..., t_{K-1}\}$.

When we continue inserting splitting indexes for $m = K$, Proposition 2 indicates that $\triangle f_*^{(K,k)}$ is constant for any $k$ because there are no $\{t_1, t_2, ..., t_{K-1}\}$ in any interval $(t_{k-1}, t_k), k \in 1, 2, ..., K$. Thus, the function $g(m)$ remains constant for any $m \geq K$. Since $g(m)$ is monotonically increasing for $m < K$, $m = K$ is the only inflection point for the function $g(m)$. Consequently, our Alg. 1 will output the true cluster number $K$.

---

**Algorithm 1** The split-and-refine algorithm for searching diagonal blocks.

---

**Input:** an ordered affinity matrix $\mathbf{W}$
**Output:** the clustering assignment $\{\mathcal{C}_k\}_{k=1}^K$

1: Initialize $\boldsymbol{\tau}^{(1)} = \{\}$
2: **for** $m = 1 : L - 1$ **do**
3:    **for** $k = 0 : (m - 1)$ **Parallelly do**
4:       a) **Split** the $k$th interval into two subsets to form the new segmentation indexes $\boldsymbol{\tau}^{(m,k)}$;
5:       b) Compute $f_*^{(m,k)} = \max\{f_k(\tau; \boldsymbol{\tau}_{-k}^{(m,k)}) : \tau_{k-1}^{(m,k)} < \tau < \tau_{k+1}^{(m,k)}\}$, and denote the maximizer as $\tau_k^*$; denote $\hat{\boldsymbol{\tau}}^{(m,k)} = (\tau_1^{(m,k)}, ..., \tau_{k-1}^{(m,k)}, \tau_k^*, \tau_{k+1}^{(m,k)}, ..., \tau_m^{(m,k)})$;
6:       c) Compute the cost reduction $\triangle f_*^{(m,k)}$ as in (2);
7:    **end for**
8:    Pick $k^* \triangleq \arg\max_k \triangle f_*^{(m,k)}$, and assign the segmentation as $\boldsymbol{\tau}^{(m+1)} = \hat{\boldsymbol{\tau}}^{(m,k^*)}$.
9:    **repeat**
10:       **for** $k = 1 : m$ **do**
11:          $\tau_k^* = \text{argmax}_{\tau_k} f_k(\tau_k)$
12:          **if** $\tau_k^{(m+1)} \neq \tau_k^*$ **then**
13:             **Refine** $\tau_k^{(m+1)} = \tau_k^*$.
14:          **end if**
15:       **end for**
16:    **until** $\{\tau_k^{(m+1)}\}_{k=1}^m$ cannot be changed
17:    Save objective function value as $g(m)$ with the segmentation $\boldsymbol{\tau}^{(m+1)}$.
18: **end for**
19: **for** $m = 2, 3, ..., L$ **do**
20:    Calculate $g_m^{''} = (g(m) - g(m-1)) - (g(m+1) - g(m))$.
21: **end for**
22: $K = \text{argmax}_{K \in 2,3,...,L-1} g_m^{''}$
23: Calculate $\{\mathcal{C}_k\}_{k=1}^K$ according to the segmentation $\boldsymbol{\tau}^{(m)}$ and the order $O$.

---

