# OpenReview forum: "Diagonalizing Affinity Matrix to Identify Clustering Structure"
_ICLR.cc/2025/Conference — Submitted to ICLR 2025_

### Official Review · Reviewer_N4Tm · 2024-10-29

**Soundness:** 2
**Presentation:** 2
**Contribution:** 2
**Rating:** 3
**Confidence:** 4

**Summary:**

The paper proposes a strategy for identifying cluster structures from an affinity matrix. This strategy can be divided into two steps: the first step permutes the affinity matrix to form a block-diagonal structure by a density-based traversal algorithm; the second step identifies the cluster structures in the permuted matrix through a split-and-merge approach.

**Strengths:**

1. Some theoretical analyses are provided.

**Weaknesses:**

1. The analysis of the algorithm's time complexity and space complexity is lacking.
2. The contribution section mentions that the strategy is "rapid," but no efficiency-related experiments are presented.
3. Although many comparison algorithms are employed, Table 1 contains numerous missing entries, making the experimental results less convincing.
4. Since deep clustering incorporates feature representation learning modules, it typically achieves better performance, especially in image applications. However, according to the tables, DAM, as a traditional algorithm, significantly outperforms various deep learning algorithms. It appears that the performance results of comparison algorithms may come from previous papers, with RGB pixels provided directly as features for algorithms like DEC, while DAM uses newly extracted features. If true, the source of performance should be noted, along with details on how the new features were extracted. More rigorously, the input for each algorithm should be standardized.
5. There is room for improvement in the paper’s writing and organization. For example, Sections 3.1 and 3.2 feel disconnected, making it difficult to understand how the former supports the latter. Additionally, the meanings of $t$, $\tau$, and $\mathcal{C}$ are somewhat confusing.
6. In the experiments, DAM's input relies on the output of BDR-B, which itself learns an approximate block-diagonal structure (not yet permuted). Therefore, I am curious how DAM would perform using Gaussian similarity as input directly, and what the results would be if BDR-B's output were paired with DAM's permutation before applying spectral clustering.
7. Since a block-diagonal structure is already obtained, directly extracting the connected components of the graph should yield cluster divisions. Hence, what is the purpose or advantage of Section 3.2? Even without permutation, Section 3.2 could still be applied, raising questions about the purpose or advantage of Section 3.1 as well.

**Questions:**

See Weaknesses.

---

### Official Review · Reviewer_u8dh · 2024-10-30

**Soundness:** 2
**Presentation:** 1
**Contribution:** 1
**Rating:** 3
**Confidence:** 4

**Summary:**

This paper proposes a novel clustering method that can directly leverage the block-diagonal form of the affinity matrix to reveal the underlying clustering structure. Specially, first, they use a traversal algorithm to establish a traversal sequence based on the affinity matrix originated from data. Second, this sequence is utilized to permute the affinity matrix to uncover the block-diagonal structure. At lats, they employ a split-and-refine to detect all diagonal blocks within the permuted affinity matrix.

The main contribution of this paper is to propose a new method to detect all diagonal blocks within the matrix.

**Strengths:**

The authors have done enough experiments.

**Weaknesses:**

1.The motivation of this paper is not clear. It is not clear why it is necessary to use the block-diagonal structure of the affinity matrix directly to reveal the underlying clustering structure.

2.The writing of the paper needs to be improved. For example, Section 2.1 does not summarize relevant work and divided it into different categories.

3.In section 4.2 of the paper, the experimental results are only presented, but not analyzed.

4.There are also many works that directly use block-diagonal structure to achieve clustering results, such as [1,2]. It is recommended to compare these algorithms and discuss them in detail.

5.It is suggested that the authors can complete the missing experimental results in Table 1 and Table 2, which may be more convincing.

[1] One-Step Multi-View Spectral Clustering.

[2] Unified one-step multi-view spectral clustering.

**Questions:**

Please see the weaknesses.

---

### Official Review · Reviewer_tcHt · 2024-11-02

**Soundness:** 3
**Presentation:** 2
**Contribution:** 2
**Rating:** 5
**Confidence:** 4

**Summary:**

In this work, the authors propose a clustering strategy based on an affinity matrix. This strategy uses a traversal algorithm to identify high-density clusters within a graph weighted by the affinity matrix, thereby establishing a traversal sequence. This sequence is then used to reorder the affinity matrix, revealing its inherent block-diagonal structure. Extensive experiments confirm the feasibility and effectiveness of the algorithm.

**Strengths:**

The motivation of the article is relatively easy to understand, and the paper is supported by a solid theoretical foundation.

**Weaknesses:**

1.The writing of the article is relatively weak, especially in Section 3.2, where the logic is somewhat lacking.

2.A framework-style diagram illustrating the detailed process of the algorithm is missing and would be beneficial.

3.Since a traversal method is used to uncover the block-diagonal structure, the computational complexity is likely quite high. Please analyze the computational complexity of the proposed algorithm.

4.In the DAM model, the number of clusters is adaptively determined, making it, to some extent, a clustering algorithm for generalized category discovery. However, many comparison algorithms require the number of clusters to be predefined, which makes this comparison somewhat unfair.

**Questions:**

See Weaknesses.

---

### Official Review · Reviewer_yy1w · 2024-11-02

**Soundness:** 2
**Presentation:** 3
**Contribution:** 2
**Rating:** 3
**Confidence:** 4

**Summary:**

This paper proposes the Diagonalizing Affinity Matrix (DAM) clustering method, which reorders the affinity matrix into a block-diagonal structure using a density-based traversal algorithm. The method then identifies clusters through a split-and-refine process, purportedly optimizing the clustering structure without needing predefined parameters.

**Strengths:**

1. The paper defines its focus on improving clustering by utilizing the block-diagonal structure of the affinity matrix, which aligns with established clustering challenges.
2. DAM does not rely on manual parameter tuning, which could theoretically simplify its application across datasets with consistent structures.

**Weaknesses:**

1. The approach essentially repurposes existing block-diagonal concepts with minor adaptations. Both the traversal and split-and-refine strategies lack substantial novelty, building primarily on established affinity-based clustering frameworks. This incremental improvement may not meet the expectations of a top-tier conference seeking fundamentally new insights or methodologies in clustering.
2. Despite claims of computational efficiency, the paper lacks any detailed analysis of DAM's scalability, especially for large datasets. The density-based traversal and iterative refinement are potentially computationally expensive, particularly when dealing with dense affinity matrices. This oversight raises concerns about the practical viability of DAM for large-scale applications. Many anchors based methods have achieved great progress.
3. The paper primarily compares DAM with older clustering methods (Very few papers in 2024, let alone comparative methods.), bypassing recent advances in deep clustering and self-supervised clustering techniques that have demonstrated state-of-the-art performance on high-dimensional and complex datasets. This selective benchmarking weakens the evidence for DAM's claimed superiority and raises questions about its actual competitiveness.

**Questions:**

1. How would DAM perform on datasets with complex, overlapping cluster structures or where the affinity matrix does not naturally exhibit a block-diagonal structure?
2. What is the computational complexity of DAM when applied to large-scale datasets, especially regarding time and memory usage?

---

### Meta-Review · Area_Chair_nVDb · 2024-12-15

**Metareview:**

All the reviewers consensually suggest rejecting the paper, and there is no author rebuttal.

**Additional Comments On Reviewer Discussion:**

NA

---

### Decision · Program_Chairs · 2025-01-22

Reject